# Using AI–ML to Augment the Capabilities of Social Media for Telehealth and Remote Patient Monitoring

**DOI:** 10.3390/healthcare11121704

**Published:** 2023-06-10

**Authors:** Ricky Leung

**Affiliations:** 1School of Public Health, University at Albany, Albany, NY 12222, USA; rleung@albany.edu or rleung@nus.edu.sg; 2Business School, National University of Singapore, Singapore 119245, Singapore

**Keywords:** AI, ML, health organizations, social media

## Abstract

Artificial intelligence (AI) and machine learning (ML) have revolutionized the way health organizations approach social media. The sheer volume of data generated through social media can be overwhelming, but AI and ML can help organizations effectively manage this information to improve telehealth, remote patient monitoring, and the well-being of individuals and communities. Previous research has revealed several trends in AI–ML adoption: First, AI can be used to enhance social media marketing. Drawing on sentiment analysis and related tools, social media is an effective way to increase brand awareness and customer engagement. Second, social media can become a very useful data collection tool when integrated with new AI–ML technologies. Using this function well requires researchers and practitioners to protect users’ privacy carefully, such as through the deployment of privacy-enhancing technologies (PETs). Third, AI–ML enables organizations to maintain a long-term relationship with stakeholders. Chatbots and related tools can increase users’ ability to receive personalized content. The review in this paper identifies research gaps in the literature. In view of these gaps, the paper proposes a conceptual framework that highlights essential components for better utilizing AI and ML. Additionally, it enables researchers and practitioners to better design social media platforms that minimize the spread of misinformation and address ethical concerns more readily. It also provides insights into the adoption of AI and ML in the context of remote patient monitoring and telehealth within social media platforms.

## 1. Introduction

The widespread use of social media has provided health organizations with new opportunities to enhance telehealth, remote patient monitoring, and operations in general [1,2]. Studies have shown that social media can be an efficient marketing tool to boost sales and revenue [3]. Health organizations such as hospitals, clinics, pharmaceutical companies, and community health centers use social media as a platform to promote their offerings, such as products and services, as well as their associations with reputable partners [4]. Additionally, the prevalence of social media use among health organizations has made it important for them to maintain a presence on these platforms through the creation of social media profiles, as it serves as a signal to the market that the organization is active and operating [5]. In other words, social media has become a vital tool for health organizations to maintain their visibility and establish a positive reputation in the market [6].

Social media has expanded the ways in which health providers can communicate with patients and other stakeholders [7]. Health providers use social media to share information about treatments and services for specific diseases, educate the public about health issues, and stay in touch with patients [8]. Additionally, doctors can use social media to promote their services and connect with potential patients [9]. On the other side, patients use social media to gain health knowledge, connect with other patients with similar health concerns, and access telemedicine and other online health services [10]. These interactions between health providers and patients also provide valuable data for health organizations to improve their competitive advantage and perform in-depth data analysis. Social media serves as a new channel for data collection and communication in the healthcare industry [11].

Thirdly, social media enables health organizations to improve relationship management with their employees, customers, and external partners [12]. Health organizations are interested in the long-term marketing and communication benefits of using social media, not just one-time or temporary benefits [13]. Social media provides a platform for the organization to connect with its stakeholders without time constraints. Patients can give a “like” or provide feedback comments to the organization whenever they remember to do so. There is no need to do these actions at a specific time. Using social media platforms, such as leaving a comment on a Facebook page or forwarding a Twitter hashtag, is perceived as more “friendly” than providing feedback or complaints in a formal manner [14]. This can also increase the health organization’s capacity to manage its social capital.

Despite the benefits mentioned above, the use of social media raises concerns such as privacy [10] and the spread of misinformation [15] among different organizational actors. There have been instances of patient data mishandling or misuse by hospitals and clinics. Additionally, some patients have obtained medical advice from unqualified sources on social media without realizing it, leading to incorrect or harmful treatments. The integration of artificial intelligence (AI) and machine learning (ML) into social media has enhanced its capabilities in positive ways, but it has also exacerbated the concerns of privacy and misinformation [16]. Many health organizations remain skeptical about the safety of AI–ML, especially when it is used in conjunction with social media.

Given these observations, this paper aims to achieve three main objectives: Firstly, the adoption of AI–ML for improving social media is increasing among health organizations. Yet, using this new technology specifically for telehealth and remote patient monitoring has been slower [17]. As such, this research initializes new directions in which AI–ML can be utilized to enhance social media in areas such as marketing, data collection, and long-term relationship management. Second, by gaining a deeper understanding of AI–ML-related research and the new technical capabilities they offer, leaders in health organizations can better evaluate whether the risks associated with social media can be mitigated by using AI–ML, and in turn, determine if investing in such technology is worthwhile for the organization. Third, this paper proposes a conceptual framework that addresses gaps in previous research and identifies critical elements for health organizations to utilize AI–ML technology effectively to enhance their social media capability. 

## 2. Social Media Marketing

Based on extensive findings, social media is a low-cost and effective marketing and telehealth tool. Strong evidence has shown that social media marketing can lead to increased brand awareness and customer engagement [18]. Social media has also been found to be an effective way for businesses to gather customer feedback and insights [19]. However, utilizing social media in business has not been without its challenges. Some studies have found that the sheer volume of data generated by social media can make it difficult for businesses to effectively analyze and use the information [20]. There are also concerns about the potential risks to customer privacy [21].

In the healthcare industry, the use of social media has been found to have numerous potential advantages. Patients have been found to use social media to research their symptoms and connect with others who have similar conditions [22]. Healthcare providers have also been found to use social media to educate the public about health issues and promote healthy behaviors [23]. However, there have also been concerns about the spread of misinformation on social media, with some studies finding that patients may seek medical advice from unqualified sources on social media [24].

The landscape on the use of AI–ML-based software, devices, and algorithms has developed rapidly in different medical areas [17]. Artificial intelligence (or AI hereafter) can be defined as a simulation of human intelligence in machines that are programmed to think, learn, and mimic human behaviors [25]. Within AI, machine learning (or ML hereafter) refers to developing algorithms and statistical models that enable machines to improve their performance with experience, using training and testing data [25]. In other words, ML is a method for teaching machines to learn from data, without being explicitly programmed.

ML has the potential to significantly change the way social media marketing is conducted [26]. Several research directions of ML are impacting social media marketing and how it is likely to continue to do so in the future. Personalization is a new capability made possible by ML [27]. ML algorithms can be used to analyze a patient user’s past behavior and use that information to tailor the content that is shown to them. This can be seen in personalized news feeds and recommendations on platforms such as Facebook and Instagram [27,28]. By showing users content that is more likely to be relevant and engaging to them, ML can help improve the effectiveness of social media marketing campaigns, including vaccination [26,27,28,29].

ML can be used to identify patterns in user data that can be used to target specific audiences [30]. For example, a social media marketing campaign for a new fitness product might use ML to target users who have shown an interest in fitness-related content in the past [31]. This can help ensure that marketing efforts reach the right audience and are more effective. ML algorithms can be used to optimize the placement and frequency of ads on social media platforms [32]. By analyzing user data, ML can help identify the best times and locations to show ads to maximize their impact. With other personalization techniques, AI–ML technology can be applied to moderate and filter content on social media platforms [33]. With a vast amount of user-generated content being posted, it can be difficult for platforms to manually review everything. AI algorithms can lead to more precise content filtering, which helps identify and remove inappropriate or malicious content, such as hate speech or spam [34].

Similarly, ML can be used to analyze the sentiment of social media posts and comments. Sentiment analysis can help companies understand how their products or services are perceived by consumers [35]. This can be useful for identifying potential issues or areas for improvement and identifying brand advocates that can be leveraged for marketing efforts. The integration of ML in social media marketing has the potential to significantly improve the effectiveness of marketing campaigns and help companies better understand and engage with their customers [36]. As ML techniques continue to advance, we will see more widespread adoption of these technologies in social media marketing. 

The above techniques may be combined to create one or more recommender systems in the health organization’s social media sites [37]. One way in which recommender systems are used on social media is through personalized news feeds [38]. For example, on Facebook, a recommender system might analyze the pages a user has liked and the posts they have engaged with in the past and use that information to show the user more posts from similar pages or topics. Similarly, on YouTube, a recommender system might analyze the videos a user has watched in the past and use that information to suggest related videos [38].

Gaps in research on social media marketing for health organizations include the need for a deeper understanding of the necessary components to enable relevant strategies and tactics, the exploration of ethical considerations surrounding targeted advertising and data privacy, the measurement of outcomes and impact on health behaviors, and the identification of best practices for engaging diverse populations. Additionally, there is a lack of research on the integration of AI–ML into social media marketing strategies, and what resources and technologies determine effectiveness and efficiency in reaching and engaging target audiences.

## 3. Social Media as a Data Collection Tool

As mentioned, health organizations now interact with patients and other stakeholders regularly through social media channels. These interactions not only improve remote patient monitoring but also provide opportunities for health organizations to collect valuable data at different levels [39]. Firstly, patient users may share information about their health, such as symptoms, diagnoses, and treatment options, on social media platforms [40,41]. This user-generated content can be used by health organizations to gain insights into public health trends, such as the spread of infectious diseases or the effectiveness of certain treatments. For example, during the recent outbreak of the COVID-19 pandemic, hospitals and public health agencies were able to use social media to identify the “hotspots” of COVID-19 occurrence and learn how different local communities were creating innovative resilience programs to cope with the pandemic [42].

Another way social media platforms collect data for health organizations is through technical tracking tools, such as cookies and pixels [43]. These tools may be used to track user behavior and preferences, which also reveal user demographics for organizations to target advertising and content. Using sophisticated analytic techniques, health organizations can also use these data to understand the online behavior of their target population, such as how they search for health-related information [44]. Additionally, social media platforms may collect data for health organizations through third-party partnerships [45]. These partnerships may involve health organizations providing social media platforms with data, such as patient data, in exchange for access to the platform’s user data. This is possible as many health organizations have now created their own internet-based platforms, allowing users to log in using their social media profile as an alternative log-in option. As a result, these users allow the health organization to access their social media identity.

The collection of data through social media can raise concerns about patients’ privacy and identity. AI–ML can be used to create privacy-enhancing technologies (PETs) that safeguard user privacy online [46]. These technologies, such as data anonymization [46,47], encryption of communications [47], and secure systems against external threats [48], can protect sensitive information such as names, addresses, and social security numbers by removing personal identifying information from data before it is shared or analyzed. This allows organizations to conduct research and analysis while still maintaining user privacy.

Anonymization of data is crucial in maintaining the privacy of individuals by ensuring that their personal information is not shared or utilized without consent. ML algorithms can be employed by health organizations to automatically identify and eliminate personal identifying information from data sets [46]. This can be achieved through training the algorithms on a large dataset, where personal identifying information has been removed manually. Additionally, AI can be used to develop methods for safeguarding privacy when sharing or analyzing data. For instance, researchers may use AI to create synthetic data that resembles real data but does not contain any personally identifying information [49]. This synthetic data can be used as a substitute for real data, allowing researchers to conduct their work without compromising the privacy of individuals.

More specifically, AI–ML can aid in the creation of encryption technologies that safeguard the privacy of communication [47]. These techniques employ mathematical formulas to encrypt data in a way that can only be deciphered by a person with the appropriate key. AI can optimize and enhance the security of these formulas, guaranteeing that communications are protected from unauthorized access. AI can also be employed to secure systems against external dangers, such as cyber-attacks [48]. Specifically, ML algorithms can be trained to identify patterns of suspicious activity and notify security teams when an attack is detected. Additionally, AI can automate the process of addressing threats, ensuring that systems are protected in real-time [50].

Depending on the context, AI can assist in designing a social media system that prioritizes privacy for health organizations. For instance, AI can be utilized to create algorithms that can automatically detect and notify users of potential privacy threats when using social media or other online platforms [48]. This can prevent unintentional disclosures of personal information and safeguard users’ privacy. Another application of AI is in conducting accurate evaluations of the possible privacy effects of new technologies and systems, helping organizations identify and reduce any potential risks. Machine Learning algorithms can be developed to perform privacy impact assessments (PIAs) to identify and minimize potential privacy hazards associated with new technologies and systems [23,51,52]. PIAs are a methodical procedure for assessing the impact of a new initiative on an individual’s privacy and are often legally required in specific industries.

There are significant gaps in research regarding the use of AI–ML to enhance social media as a data collection tool for health organizations. Firstly, there is a lack of comprehensive studies exploring the accuracy and reliability of various AI modalities in extracting health-related information from social media platforms. Additionally, ethical considerations, such as privacy protection and informed consent, need to be thoroughly examined to ensure the responsible use of collected data. Furthermore, there is a need for research on integrating diverse data sources and developing robust methodologies to effectively interpret and analyze the vast amount of unstructured data available on social media platforms. Specific techniques such as natural language processing (NLP) and sentiment analysis need to be studied more carefully.

## 4. Long-Term Relationship Management

AI–ML also enables organizations to improve long-term relationship management with various stakeholders. This relationship is important to improve the probability of success in both telehealth and remote patient monitoring. First, social media can collect user data by creating AI-enabled chatbots [53]. Chatbots are computer programs designed to simulate conversation with human users, especially over the Internet. They can be used on social media platforms to provide customer service, answer questions, and even suggest products or services. Research showed that these chatbots provide useful therapeutic value to psychiatric patients [53]. Health organizations can collect these conversation data to better understand the preferences of their customers.

More generally, chatbots can automate repetitive and time-consuming tasks related to stakeholder engagement, such as answering frequently asked questions, scheduling appointments, and providing information [1,54]. This can free up resources for organizations to focus on more strategic relationship-building activities. In addition, chatbots can be available 24/7 to engage with stakeholders, providing them with prompt and personalized responses at any time [55]. This can help improve stakeholders’ experience and make it more convenient for them to interact with the organization.

Using chatbots and related technologies, AI–ML allows organizations to analyze large amounts of data to identify potential privacy risks [56]. For example, ML algorithms can be used to scan data sets for personal identifying information that may have been inadvertently included. ML-driven data analysis enables health organizations to evaluate the likelihood and potential impact of identified privacy risks. This can help the organizations prioritize which risks should be addressed first [57]. Results from ML-driven data analysis also improve privacy controls [58]. AI can be used to identify and recommend privacy controls that can be implemented to mitigate identified risks. As mentioned, AI can lead to the use of encryption [46,47,48] or anonymization techniques [46,47,48] to protect personal data. Drawing on these techniques, it is possible to build automated AI technologies to enable continuous monitoring [59,60,61]. With these technologies, the health organization can continuously monitor systems and technologies for privacy risks, alerting organizations to any potential issues and helping them to take timely action to address them.

In the longer run, AI can increase the effectiveness of user education in social media usage as well [59]. AI can be used to help educate users about privacy risks and best practices for protecting their personal information online. In many organizations, AI is already applied for training and education in a variety of contexts, such as nursing [61]. In addition, AI can be used to create personalized learning experiences that adapt to the needs and preferences of individual users. This can be particularly useful for learning privacy risks, where AI-powered tutors can provide interactive educational content, such as simulations and games, that help users learn important concepts in a more engaging and effective way [62]. This type of content can be particularly useful for helping users learn technical subjects in terms of privacy, as it allows them to explore and experiment in a safe and controlled environment.

To enhance personalized recommendations for further learning, an AI system might track a user’s progress through an online course and suggest additional resources or activities that would be most helpful for improving their skills [63]. In this sense, AI has the potential to revolutionize the way we think about user education by providing personalized, interactive, and data-driven learning experiences. By leveraging the power of AI, educators can create more effective and engaging educational content, and help users learn more efficiently and effectively [63].

The research gaps in using AI–ML to enhance social media for long-term relationship management and remote patient monitoring are notable. Firstly, there is a lack of studies examining the ethical and privacy concerns surrounding AI–ML implementation in these areas. Additionally, limited research exists on the integration of AI–ML algorithms with existing healthcare systems, hindering seamless data exchange and interoperability. It is particularly important to consider if new forms of long-term relationship management may arise due to the use of AI–ML in social media platforms. Furthermore, there is a need for empirical studies to evaluate the effectiveness, scalability, and cost-efficiency of AI–ML technologies in long-term relationship management and remote patient monitoring within diverse healthcare settings. Addressing these gaps is crucial to ensure the responsible and effective utilization of AI–ML in enhancing these aspects of healthcare.

## 5. A Conceptual Framework of AI–ML Enhancement of Social Media

Given the gaps identified above, this section proposes a new conceptual framework that explores the potential of AI–ML in social media for health organizations. Figure 1 provides a visualization of the proposed framework. The framework identifies critical elements that enable researchers and practitioners to harness the power of AI–ML to enhance social media marketing, data collection, and long-term patient monitoring. Practically, the framework not only addresses gaps in previous research but can lead to more effective health campaigns, improved public health strategies, and enhanced patient care.

It is noteworthy to highlight the major components of this framework. First and foremost, by considering various social media platforms, such as Facebook, Twitter, or Instagram [1], the framework accommodates the diverse communication channels patients and healthcare providers utilize. Different data collection modes, such as surveys, text analysis, or user-generated content, ensure a comprehensive understanding of patient experiences and behaviors [11]. Incorporating a wide range of data features, such as demographics, sentiment, or health-related discussions, enables a deeper analysis of patient needs and preferences.

The above features provide useful “learning materials” for various ML algorithms, including supervised, unsupervised and reinforcement learning [24]. The integration of AI–ML modalities, such as natural language processing, machine learning algorithms, or predictive modeling, allows for efficient data processing, pattern recognition, and personalized interventions. Long-term relationship management is emphasized, enabling continuous patient engagement, follow-ups, and support through social media channels.

Ultimately, the proposed framework aims to enhance remote patient monitoring, fostering proactive healthcare interventions, early detection of health issues, and personalized care. By addressing these research gaps, the framework lays the foundation for a comprehensive and effective AI–ML approach to optimizing social media for long-term relationship management and remote patient monitoring in the context of telehealth.

To summarize, the proposed framework has several distinctive elements. Firstly, AI–ML enables targeted and personalized social media marketing, reaching specific demographics with relevant health information, fostering engagement, behavior change, and improving health outcomes. Secondly, AI–ML transforms data collection by analyzing vast social media data, uncovering insights on public health trends, sentiment analysis, and early disease detection. These insights inform strategies, identify emerging health issues, and enhance surveillance systems. Lastly, AI–ML greatly improves long-term patient monitoring by tracking and analyzing social media activities and identifying patterns and markers of health condition changes. Proactive interventions and timely support can be provided by health organizations, preventing adverse events. This also allows for more transparency in revealing ethical and privacy issues such that appropriate policies may be formulated more readily. 

Overall, the adoption of this framework can optimize social media’s potential in marketing, data collection, and long-term patient monitoring, enhancing healthcare outcomes. The expectation is that the more a health organization can adopt the framework, the more efficiency it can achieve. Yet, before a full adoption is possible, the organization may consider a partial adoption that combines traditional long-term relationship management practice and selective use of AI–ML-enabled social media platforms. Health organizations that elect not to adopt AI–ML can miss significant opportunities in this new terrain.

## 6. Conclusions

This paper reviews several studies regarding the application of AI–ML in enhancing the utilization of social media by health organizations. Subsequently, the paper proposes a conceptual framework designed to more effectively utilize AI–ML to improve social media. Social media plays a crucial role in marketing and data gathering, and its effective use can impact the organization’s reputation, curb the dissemination of false information, and enhance relationship management. These are important elements in telehealth and remote patient monitoring. In today’s competitive landscape, the ability to effectively use social media also determines whether an organization can gain and maintain a competitive edge, as well as develop dynamic capabilities. At the individual level, the ability to use social media as a reliable means of communication with customers has become a crucial tool for achieving leadership in any health field. Table 1 summarizes the advantages of social media for health organizations, the shortcomings of using social media, and potential enhancements of adopting AI–ML.

The use of social media raises privacy concerns, but AI-ML algorithms can also enhance privacy technologies, data collection, and long-term relationship management through the use of 24/7 chatbots. However, utilizing AI–ML in social media also presents challenges, such as instances of chatbots and other AI systems displaying racist or sexist behavior. To take advantage of these new capabilities, companies must be transparent about their use of AI on social media and take measures to ensure their algorithms are fair and unbiased.

The challenges of using AI-ML in social media include the potential for bias and discrimination in predictions, lack of transparency and comprehension for users, and potential for user dependency on the technology. To mitigate these challenges, organizations should be transparent, reduce bias, prioritize user privacy, and continuously evaluate and improve their AI systems to align with ethical standards.

As AI-ML becomes more prevalent in social media, there is a danger of users becoming too reliant on technology and losing the ability to assess information critically. To address these challenges, organizations should be transparent about their use of AI-ML, take steps to eliminate bias, prioritize user privacy and continuously evaluate and improve their systems to ensure they conform to ethical standards.

To sum up, the incorporation of AI-ML in social media has the potential to enhance the user experience and make it simpler for companies to interact with their customers. However, practitioners in health organizations should weigh the potential consequences and ensure that the use of AI-ML is responsible and adheres to ethical standards.

## Figures and Tables

**Figure 1 healthcare-11-01704-f001:**
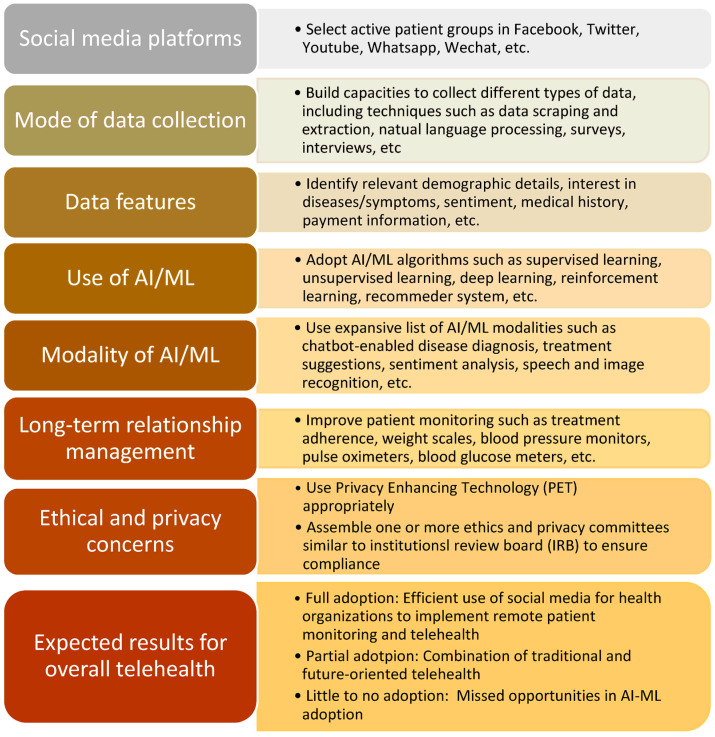
A conceptual diagram that identifies essential components in AI–ML adoption to improve social media for health organizations.

**Table 1 healthcare-11-01704-t001:** Potential effects of using AI–ML to enhance social media for health organizations.

	Social Media Advantages	Shortcomings of Social Media	AI–ML Enhancements
Social Media Marketing	Low-cost tool for advertisement Promote healthy behaviors	Difficulty in targeting specific patient groups Information overflow	Personalization and audience targeting Optimizing frequency and placement of ads
Data Analysis	Predictive analytics Tracking of public health behaviors	Privacy related to personal health data Privacy of patients’ communication	Privacy-enhancing technologies (PETs) Anonymization and encryption
Long-term Relationship Management	Online existence Engagement of customers	Labor cost to provide customer service Repetitive tasks require enormous administrative costs	Chatbots provide 24/7 customer service Automation of customer responses and repetitive tasks

## Data Availability

Not applicable.

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
