# Peer review of "Using AI–ML to Augment the Capabilities of Social Media for Telehealth and Remote Patient Monitoring"

_healthcare, 2023, doi:10.3390/healthcare11121704_

Round 1

Reviewer 1 Report

Dear Author,

The manuscript for the concept paper is interesting however the manuscript can be improved in several ways. Here are some of the recommendations.

Minor comments:

1.     AI/ML is a broad term. It is recommended that author expand upon the modality of AI/ML that would be most suitable in chatbots, sentiment analysis, telehealth and remote patient monitoring.

2.     In introduction section, line 61-65 author could mention how use of AI/ML is adopted in other area of healthcare settings, and the adoption is increasing however this is not a case in telehealth and patient monitoring.

It is recommended that author cite this paper https://www.medrxiv.org/content/10.1101/2022.12.07.22283216v2.full-text , that provides updated landscape on use of AI/Ml based software/devices/algorithms in different medical areas. Figure 3 provides the trend in use of AI in different medical specialty. Author could then mention the use of AI/ML in telehealth and patient monitoring needs consideration.

Major Comment:

1.     It is recommended that author include a flow chart/diagram that summarizes social media platforms, mode of data collection, data features that needs to be collected, use of AI/ML, modality of AI/ML, methods in long term patient relationship management, and overall telehealth.

In another words, it is recommended that author summarize the important key points of the whole paper in schematic diagram.

Minor spellcheck required.

Author Response

Thank you very much for reviewing my manuscript entitled “Using AI-ML to augment the capabilities of social media for telehealth and remote patient monitoring”. I have revised the manuscript. Below are my specific responses to your comments/suggestions:

Point-to-point responses:

  1. I have expanded the modality of AI/ML in the paper which better encompasses technologies such as chatbots, sentiment analysis, telehealth, etc.
  2. I mentioned that AI/ML adoption is increasing but not as much in telehealth and patient monitoring in lines 71-73. I reviewed and cited the paper by Joshi et al. (as per Reviewer 1’s suggestion) and included in the reference list (no. 17).
  3. I created a diagram that summarizes the flow of the paper, and provide a visualization of how different conceptual elements fit together.

Reviewer 2 Report

Artificial intelligence and machine learning are changing how healthcare organizations use social media. These technologies can manage the vast amount of data generated by social media, improving telehealth, remote patient monitoring, and the overall well-being of society. One application of AI is to improve social media marketing through sentiment analysis, which increases brand awareness and customer engagement. Social media can also be used as a data collection tool, but it requires careful privacy protection through privacy-enhancing technologies. AI-ML also enables organizations to maintain long-term relationships with stakeholders using chatbots and personalized content. While these technologies have great potential, it is necessary to consider their potential consequences and design them to prevent the spread of misinformation carefully.

Dear Author, 

Overall, the article is well written. However, the main idea presented in the article seems more in line with a literature review than a new contribution. Therefore, I suggest the authors consider submitting their work as a literature review.

Dear Authors, 

Some minor editing of the English language is required.

Author Response

Thank you very much for reviewing my manuscript entitled “Using AI-ML to augment the capabilities of social media for telehealth and remote patient monitoring”. I have revised the manuscript. Below are my specific responses to your comments/suggestions:

Point-to-point responses:

  1. While this paper might be expanded and rewritten as a review, we think the main goal is to contribute to conceptual development. So, we proposed a framework has been proposed.
  2. Also, in the abstract, we rewrite lines 19-24 in the abstract to better position this paper as a concept paper. Additionally, we added lines 79-81, added a new Section 5 to discuss the conceptual framework, and created Figure 1 to identify essential components in AI-ML adoption.

Reviewer 3 Report

The author applied the AI-ML to augment the capabilities of social media for telehealth and remote patient monitoring efficiently. The work has merits. There are some suggestions in sake of enhanced paper quality as below:
1. What parameters the proposed methodology is better as compared to existing techniques ?
2. What is the overall analysis of the proposed technique?
3. Add more broad information with regard to the Start of the Background, Problem Definition, Scope and other related information.
4. In related work, all the papers should be elaborated with what is proposed, what is the novelty and what experimental results are there.
5. In the end of related work section, highlight in 9-15 lines what overall technical gaps are observed in existing techniques that led to the design of the proposed methodology.
6. Add the details of the Methods which are used as base to design and develop the proposed methodology in section 4.
7. Add the Steps of working, Algorithm and flowchart of the proposed methodology.
8. System Model is missing and under system model, only mathematical modeling of the proposed technique must be done.
9. Experimental Setup is missing, and then give Experimental Parameters.
10. Compare the proposed technique with existing techniques and even add the Tables with Data Values and the Graphs.
11. Make the section 5 more detailed and even add what is the proposed methodology and what experimental parameters are there and what conclusions are there. 12. Addition of future scope is also recommended. And before conclusion add some Case study based discussion.
13. Kindly improve the writing of your manuscript in order to facilitate its potential readers' understanding.

Author Response

Thank you very much for reviewing my manuscript entitled “Using AI-ML to augment the capabilities of social media for telehealth and remote patient monitoring”. I have revised the manuscript. Below are my specific responses to your comments/suggestions:

Point-to-point responses:

  1. We reviewed selected studies in the field, identified research gaps, and proposed a conceptual framework.
  2. Same as above.
  3. We added more discussions throughout the revised manuscript, such as lines 19-24, 71-73, and 79-81.
  4. We identified research gaps and discussed them in lines 143-150, 214-223, 272-282.
  5. We discussed these in the new section 5, which detailed the conceptual framework we proposed.

6-12. This paper is mainly a concept paper, so experimental methodology is out of the scope.

  1. We rewrote the paper and proofread it.

Reviewer 4 Report

This paper is a sort of review, however, it’s too short for being a review paper. More topics should be investigated. This should be reflected on your reference list as only 39% of your references are from the last 5 years. References should be updated to cover more topics from recent literature.

The author is not clear about the purpose of his work, this should be clear in the abstract. 

A paragraph should be added to outline the structure of this paper.

The whole text is just a big introduction, the paper has a low correlation to telehealth. 

I strongly suggest modify the paper accordingly. 

Author Response

Thank you very much for reviewing my manuscript entitled “Using AI-ML to augment the capabilities of social media for telehealth and remote patient monitoring”. I have revised the manuscript. Below are my specific responses to your comments/suggestions:

Point-to-point responses:

  1. We added a new section 5 which discussed the conceptual framework we proposed in detail.
  2. We updated our reference list to cover more research from the literature.
  3. We added more discussion in the abstract, clarified the paper structure, and specified how the paper is related to telehealth and remote monitoring.

Round 2

Reviewer 1 Report

The comments are addressed.

Minor editing of English language required

Reviewer 2 Report

Thank you for addressing the proposed comments. 

Moderate English language revision is required